Effects of 3021 meal replacement powder protect NAFLD via suppressing the ERS, oxidative stress and inflammatory responses

Xie Qi 1
Gao Shuqing 1
Li Yuanjudi 2
Xi Weifang 3
Dong Zhiyun 2
Li Zengning 4 zengningli@hebmu.edu.cn
Lei Min 5 leimin8@hebmu.edu.cn
1 The Forth Hospital of Hebei Medical University , Shijiazhuang , China
2 Shenzhen Anxintang Biotechnology Co., Ltd , Shenzhen , China
3 Xinchen Biotechnology (Guandong) Company Limited , Dongguan , China
4 The First Hospital of Hebei Medical University, Hebei Province Key Laboratory of Nutrition and Health , Shijiazhuang , China
5 The Third Hospital of Hebei Medical University , Shijiazhuang , China
Stochaj Ursula
Electronic publication date: 2023 Oct 16
Publication date: 2023
Volume: 11
Electronic Location ID: e16154
Received 2023 Mar 20; Accepted 2023 Aug 31
Copyright: © 2023 Xie et al.
Copyright year: 2023
Copyright holder: Xie et al.
License: This is an open access article distributed under the terms of the Creative Commons Attribution License, which permits unrestricted use, distribution, reproduction and adaptation in any medium and for any purpose provided that it is properly attributed. For attribution, the original author(s), title, publication source (PeerJ) and either DOI or URL of the article must be cited.
License URL: https://creativecommons.org/licenses/by/4.0/

Keywords: 3021 meal replacement powder, miR-26a, NAFLD, Obesity, ERS, Oxidative stress, Inflammation

Funding: The authors received no funding for this work.

==============================
Objective

To explore the specific protective mechanism of 3021 meal replacement powder (MRP) against non-alcoholic fatty liver disease (NAFLD).

Materials and Methods

C57BL/6J male mice were divided into four groups: control group, 3021 MRP group, model group and test group. The lipid accumulation and endoplasmic reticulum stress (ERS)-related proteins in hepatocytes of mice were detected by hematoxylin-eosin (HE) staining, oil red O staining and Western blotting.

Results

The expressions of GRP78, GRP94, p-PERK and p-IRE1α were significantly inhibited in test group compared with those in model group. The protein expressions of p-NF-κB, p-JNK, IL-1β, IL-18 and NOX4 in test group were also significantly lower than those in model group. In vivo and in vitro experiments revealed that the body weight and lipid droplet content, and the expressions of ERS-related proteins (including BIP and XBP-1) in liver tissues all significantly declined in model group compared with those in 3021 MRP group.

Conclusion

In conclusion, 3021 MRP can greatly reduce lipid accumulation by inhibiting ERS, oxidative stress and inflammatory response in NAFLD.

Introduction

Obesity has emerged as a major public health concern with the improvement in living standards and changes in lifestyle (Mayer et al., 2009; Hu et al., 2018; Li et al., 2023). As a chronic metabolic disease caused by bad dietary habits and a sedentary lifestyle, obesity can also increase the risk of hyperlipidemia (Tóth et al., 2021), type 2 diabetes mellitus (Yang et al., 2021) and cardio-cerebrovascular diseases (Dos Santos et al., 2021), all of which fundamentally result from lipid metabolic imbalance in the liver. Excess carbohydrates can be converted in the liver into triglyceride (TG), which is then transferred and stored in adipose tissues, ultimately resulting in obesity (Ooi et al., 2021). The liver acts as an important organ in lipid synthesis, and it is prone to massive lipid accumulation when lipid synthesis or metabolic disorders occur. Non-alcoholic fatty liver disease (NAFLD) is caused by excessive fat accumulation in the liver, which can further lead to severe diseases, such as non-alcoholic steatohepatitis (NASH) (Clugston, 2020; Daniels et al., 2020).

Inflammation of adipose tissues is a major contributor to the development of various obesity-related diseases. In addition, endoplasmic reticulum stress (ERS) may be a core mechanism of obesity, chronic inflammation and peripheral insulin resistance (Tirosh et al., 2021). ERS is defined as a state where the homeostasis of ER is disrupted by stress factors like hypoxia, leading to protein misfolding or over-synthesis, and misfolded protein accumulation in ER (Ferraz-Bannitz et al., 2021). Generally, unfolded protein response (UPR), one of the most important states of ERS, typically indicates the occurrence of ERS (Fung & Liu, 2014). Recently, ERS and oxidative stress (OS) have been shown to play significant roles in the development into hepatic steatosis and the subsequent production of reactive oxygen species (ROS) (He et al., 2021; Cho et al., 2021). Therefore, it is speculated that appropriate intervention in the ERS and ROS may ameliorate lipid metabolism in the liver, thereby reducing the risk of obesity-related diseases. Moreover, both the ERS and ROS are powerful promotors for inflammatory responses, which are responsible for lipid accumulation (Teunis, Nieuwdorp & Hanssen, 2022; Cobbold, Patel & Taylor-Robinson, 2012).

3021 meal replacement powder (MRP) is a single or composite brewing powder made from cereals, beans and tubers supplemented by edible parts such as roots, stems and fruits of other plants (including soybean dietary fiber powder, wheat fiber, white kidney beans, collagen, soy protein isolate, skim milk powder, whole-wheat flour, wheat flour, gordon euryale seeds, and chia seeds). MRP can regulate glucolipid metabolism, increase satiety and reduce calorie intake without affecting the intake of proteins, vitamins, minerals and other nutrients in the body, thereby leading to weight loss, which may exert a positive effect on various obesity-related diseases (Wang et al., 2021; Ueoka et al., 2022; Ahmed et al., 2018; Creus et al., 2016). However, the specific mechanism of MRP in weight loss has been rarely explored in studies and clinical trials. In this study, therefore, obesity mouse models were established by a high-fat diet (HFD) to investigate the effects of 3021 MRP on ERS and ROS in obese mice and its mechanism.

Materials and Methods

Cell culture

Hepatocytes were harvested from mice and cultured in high-glucose DMEM containing 10% fetal bovine serum, 100 U/mL penicillin and 100 μg/mL streptomycin in a 5% CO2 incubator at 37 °C. Then hepatocytes in the logarithmic growth phase were selected, inoculated into a 96-well plate (1 × 106 cells/mL), and rinsed once with PBS after adherence. Afterwards, they were added with 1.2 mm non-esterified fatty acids (NEFAs) or/and 3021 MRP extract and divided into control group (treated with control mouse serum), 3021 MRP group (administrated with 10% 3021 MRP containing mouse serum and NEFAs), and model group (treated with control mouse serum and NEFAs added with 0.06–60 nM thapsigargin for 24 h or without thapsigargin) (derived from plant poison carrots, thapsigargin is a sesquiterpene lactone that permeates cells. Studies have shown that thapsigargin excretes intracellular Ca2+, resulting in rapidly elevated concentrations of cytosolic Ca2+ due to specific inhibition of endoplasmic reticulum Ca2+-ATPase, which is not involved in the hydrolysis of inositol phospholipid or protein kinase C (Jackson et al., 1988; Thastrup et al., 1990; Zou et al., 2015; Zieminska et al., 2014). In this study, thapsigargin was used to disrupt calcium ion balance to induce ERS).

RAW264.7 cells were stimulated with NEFAs, treated with 10% 3021 MRP containing mouse serum and control mouse serum, respectively, and then treated with thapsigargin. Afterward, RAW264.7 cells were co-cultured with HepG2 cells, with the corresponding controls established.

Establishment of obesity mouse models

C57BL/6J mice aged 180–250 days were purchased from the Laboratory Animal Center of Hebei Medical University, and obesity mouse models were established by 60% kcal HFD purchased from the feed company. The mice were housed in an SPF environment at 18–25 °C for 1 week. Then they were weighed, numbered according to body weight and cage number, and randomly divided into control group (normal feed and normal saline), model group (HFD and normal saline) and test group (HFD and 3021 MRP), with 10 mice in each group. After 5 weeks, the tests were performed. The body weight of mice in each group was measured before sampling. The animal use protocol had been reviewed and approved by the Institutional Animal Care and Use Committee of The Fourth Hospital of Hebei Medical University (No. IACU-4th Hos Hebmu-2020017). Mice were euthanized with 20% isoflurane in the shortest time to reduce the pain, distress, anxiety, or fear.

Hematoxylin-eosin (HE) and oil red O staining

The injury of liver tissues fixed with 4% paraformaldehyde was observed by HE staining, and the lipid droplets in the liver were observed by oil red O staining. The images were captured and analyzed under a fluorescence microscope (NI; Nikon, Tokyo, Japan).

Western blotting

The protein was extracted from cells and tissues according to the manufacturer’s instructions, and its concentration was measured using BCA kits (Beyotime, Shanghai, China). Then 20 μg of proteins were loaded for separation, blocked with 5% skim milk powder for 2 h, and incubated with primary antibodies against GRP78, GRP94, p-PERK, p-IRE1α, p-NF-κB, p-JNK, BIP, XBP-1, IL-1β, IL-6, IL-8 and GAPDH (1:1,000) at 4 °C overnight on a horizontal shaking incubator. After the membrane was washed, the proteins were incubated with the corresponding secondary antibodies (1:5,000) at room temperature for 2 h on the horizontal shaking incubator, followed by development with ECL and photography using a gel imaging system. The assay was repeated three times. The relative protein expression was expressed as the ratio of the gray value of the protein bands to that of the corresponding internal reference measured by ImageJ.

Statistical analysis

SPSS 23.0 software was used for statistical analysis. Measurement data were expressed as mean ± standard deviation (χ¯±s) and compared by one-way analysis of variance among groups, and by LSD test between two groups. Whether or not the data were consistent with the assumptions of statistical methods was assessed. If not, the data were then excluded. p < 0.05 was considered statistically significant.

Results

3021 MRP could significantly inhibit lipid accumulation in hepatocytes in vitro

It is well known that lipid accumulation precedes the development of NAFLD (Simon et al., 2019). In this study, lipid accumulation and lipid toxicity in hepatocytes could be induced by NEFAs. As shown by oil red O staining and microscopic observation, lipid droplets were stained to varying degrees in each group, the cell morphology had significant differences between control group and model group, and a large number of cells in model group were stained red, suggesting lipid accumulation in cells in model group. Compared with model group, the lipid droplet content in hepatocytes significantly declined in test group (Fig. 1), indicating that 3021 MRP can significantly reduce the lipid accumulation caused by NEFAs in hepatocytes.

Figure 1 3021 MRP could significantly inhibit lipid accumulation in hepatocytes in vitro.

(A) Microscopic observation of hepatocytes in bright fields; (B) oil red O staining results of hepatocytes.

3021 MRP could repress ERS, and relieve the NEFA-induced inflammatory response of macrophages

ERS and ROS caused by mitochondrial dysfunction usually lead to inflammatory responses in RAW264.7 cells, resulting in lipid accumulation (Li et al., 2022). The protein expressions of GRP78, GRP94, p-PERK, p-IRE1α, p-NF-κB, p-JNK, IL-1β and IL-8, as well as t-PERK, t-IRE1α, t-NF-κB, t-JNK and NOX4, were significantly lower in test group than those in model group. The above findings suggested that 3021 MRP can significantly ameliorate the abnormal activation of ERS and ROS, and impede the NEFA-induced inflammatory response of macrophages (Figs. 2A–2C, n = 8 per group).

Figure 2 3021 MRP could repress ERS and relieve the NEFA-induced inflammatory response of macrophages.

(A) Statistics of protein bands and relative protein expressions of GRP78, GRP94, p-PERK, p-IRE1α, p-NF-κB, p-JNK, t-PERK, t-IRE1α, t-NF-κB, t-JNK in RAW264.7 cells; (B) statistics of protein bands and relative protein expressions of IL-1β and IL-8 in RAW264.7 cells; (C) statistics of protein bands and relative protein expression of NOX4 in RAW264.7 cells. **p < 0.01, ns p < 0.05.

3021 MRP could inhibit the HFD-induced weight gain of mice, and ameliorate the lipid accumulation in liver tissues of mice

The mean body weight of HFD-fed mice was the largest in model group, and it was smaller in 3021 MRP group than that in model group but larger than that in control group. It was observed by HE staining that the hepatocytes in the hepatic lobules of mice were arranged normally, and the structure of hepatic lobules was intact and clear in control group. In model group, many lipid droplets and severe lipid accumulation were found in the liver, and there were obvious pathological changes such as steatosis, ballooning degeneration and inflammatory cell infiltration in hepatocytes. In test group, the lipid droplet content in the liver significantly declined compared with that in model group, and the lipid accumulation, hepatic steatosis, ballooning degeneration and inflammatory cell infiltration were ameliorated to a certain extent after intervention with 3021 MRP. The results of oil red O staining showed that the lipid droplet content in model group was significantly higher than that in control group, while it was significantly lower in test group than that in model group (Fig. 3). These results demonstrated that 3021 MRP can suppress the HFD-induced weight gain of mice, and significantly reduce the lipid accumulation in liver tissues.

Figure 3 3021 MRP could inhibit the HFD-induced weight gain of mice, and ameliorate the lipid accumulation in liver tissues of mice.

(A) Statistics of body weight of mice in each group; (B) HE staining results of liver tissues; (C) results of oil red O staining of liver tissues and analysis using Image Pro Plus 6.0. **p < 0.01, ns p < 0.05.

3021 MRP could suppress ERS and relieve the HFD-induced inflammatory response in liver tissues of mice

The protein expressions of p-PERK, p-IRE1α, p-NF-κB, p-JNK, IL-1β, IL-6 and IL-8 in liver tissues of HFD-fed mice significantly increased compared with those in control group. The protein expressions of ERS signaling pathway-related factors GRP78, GRP94, p-PERK, p-IRE1α, BIP and XBP-1, and the total protein expressions of PERK, IRE1α and the OS marker NOX4 in liver tissues were significantly lower in 3021 MRP group than those in model group, suggesting that 3021 MRP can suppress ERS. Meanwhile, it was found that the protein expressions of p-NF-κB, p-JNK, IL-1β and IL-8 as well as t-NF-κB in liver tissues in test group significantly declined compared with those in model group (Figs. 4A–4C, n = 4 per group). It can be seen that 3021 MRP can suppress ERS and ROS, and relieve the HFD-induced inflammatory response in liver tissues of mice.

Figure 4 3021 MRP could suppress ERS and relieve the HFD-induced inflammatory response in liver tissues of mice.

(A) Statistics of protein bands and relative protein expressions of GRP78, GRP94, p-PERK, p-IRE1α, p-NF-κB, p-JNK, t-PERK, t-IRE1α, t-NF-κB, t-JNK, BIP and XBP-1 in liver tissue; (B) statistics of protein bands and relative protein expressions of IL-1β and IL-8 in liver tissues; (C) statistics of protein bands and relative protein expression of NOX4 in liver tissues. **p < 0.01, ns p < 0.05.

3021 MRP could ameliorate the abnormal activation of ERS and inhibit the NAFLD-induced inflammatory response in hepatocytes

In HepG2 cells, the relative protein expressions of GRP78, p-PERK, p-NF-κB and IL-1β significantly declined in 3021 MRP group compared with those in control group. However, they significantly increased in control group and 3021 MRP group after the addition of thapsigargin, without significant differences between the two groups (Fig. 5). The results indicated that 3021 MRP can ameliorate the abnormal activation of ERS and inhibit the NAFLD-induced inflammatory response in hepatocytes (Fig. 6).

Figure 5 3021 MRP could ameliorate the abnormal activation of ERS and inhibit the NAFLD-induced inflammatory response in hepatocytes.

(A) Statistics of protein bands of GRP78, p-PERK, p-NF-κB and IL-1β in hepatocytes; (B) relative protein expressions of GRP78, p-PERK, p-NF-κB and IL-1β in hepatocytes. **p < 0.01, ns p < 0.05.

Figure 6 3021 MRP could significantly ameliorate the abnormal activation of ERS and ROS, and inhibit the NEFA-induced inflammatory response in hepatocytes in vitro and HFD-induced inflammatory response in liver tissues of mice.

Discussion

Obesity is a chronic metabolic disease primarily caused by lipid metabolic imbalance in the liver (World Health Organization, 2000; Zhang et al., 2021). Excess carbohydrate is converted into TG in the liver and transferred and stored in adipose tissues, thus resulting in obesity (Yang & Colditz, 2015). The liver is an important organ for lipid synthesis, which is prone to massive lipid accumulation in the case of lipid synthesis or metabolic disorders (Hashem, Khalouf & Acosta, 2021). It is well known that lipid accumulation is a precursor of NAFLD (Simon et al., 2019). In this study, the results showed that 3021 MRP could greatly reduce the NEFA-induced lipid accumulation and lipid toxicity in hepatocytes and liver tissues in vitro, restrain HFD-induced weight gain of mice (Fig. 1), and ameliorate NEFA-induced inflammatory response in macrophages in vitro and HFD-induced inflammatory response in mouse liver tissues through inhibiting ERS and ROS (Fig. 2).

ER, a complex cellular component widely present in eukaryotic cells and the main site of protein synthesis and processing (Pachikov et al., 2021), contains enzymes involved in lipid metabolism. ERS will be induced by abnormal accumulation of unfolded and/or misfolded proteins in ER, triggering UPR to maintain ER homeostasis (Huang, Xing & Liu, 2019). Moreover, ERS is a core mechanism of obesity, inflammation and metabolic disorders. UPR, a type of ERS, is implicated in the regulation of lipid synthesis, decomposition and transduction (Yu et al., 2017). The IRE1α signaling pathway, one of the three UPR signaling pathways, has the closest correlation with ERS and liver lipid metabolism, which is involved in regulating the synthesis of TG and the assembly and secretion of very low-density lipoproteins. Under physiological conditions, IRE1α is exposed to the domains in the ER cavity and is in an inactive state (Huang, Xing & Liu, 2019). In the case of ERS, the IRE1α signaling pathway will be activated, so that the downstream XBP-1 protein is converted into a stable and transcriptionally active transcription factor. When ERS occurs in obesity, XBP-1 can up-regulate the expression of response genes, promote the synthesis of TG and lead to the accumulation of TG in the liver (Jiang et al., 2016; Girona et al., 2019). In this study, 3021 MRP down-regulated the protein expressions of p-IRE1α, BIP and XBP-1 in mouse liver tissues in model group, and inhibited ERS, thereby promoting lipid metabolism and alleviating obesity (Teunis, Nieuwdorp & Hanssen, 2022). These findings were also supported by the fact that 3021 MRP reduced the HFD-induced weight gain of mice (Fig. 3). Nicotinamide adenine dinucleotide phosphate (NADPH) oxidase is a protein family, of which NOX4 is the most consistently associated with the ER (Zeeshan et al., 2016). NOX4/p22phox uses NADH or NADPH (as an electron donor) for oxygen reduction to superoxide anion (Bedard, Lardy & Krause, 2007). As a result, ROS is generated and then ER calcium is released to mitochondria, producing ATP and mitochondrial ROS, which in turn activates ERS-induced calcium release, thus creating a vicious cycle (Lin et al., 2019; Chaudhari et al., 2014). ERS and OS finally induce death or apoptosis of hepatocytes by overexpressing CHOP and amplifying ROS accumulation, and ERS is associated with inflammation (as NF-κB and NLRP3 are the inflammatory markers) (Zeeshan et al., 2016). Therefore, anti-ERS or -OS drugs are beneficial for the treatment of NAFLD.

Moreover, ERS can promote the activation of Nrf2, JNK, NF-κβ and CHOP, and actively takes part in inflammation and OS, thus inducing the progression of NAFLD (Lebeaupin et al., 2018; Zhang et al., 2014). Studies have shown that the dietary fiber and other phytonutrients contained in 3021 MRP can strengthen beneficial gut microbiota and improve ERS (Zou & Qi, 2020; Wang et al., 2020; Jiang & Li, 2022; Bibi et al., 2017). In this study, the results of Western blotting and immunofluorescence assay revealed that the expressions of GRP78, GRP94, p-PERK and p-IRE1α (generally considered ERS-related biomarkers) were significantly inhibited by 3021 MRP. Besides, it was also discovered that the protein expressions of p-NF-κB, p-JNK, IL-1β and IL-8 significantly declined in hepatocytes and liver tissues in test group compared with those in model group (Fig. 4). A large amount of ROS was produced after treatment with NEFAs (considered the main phenomenon after OS activation), while NEFA-induced production of ROS in hepatocytes was inhibited after intervention with 3021 MRP. RAW264.7 cells were stimulated with NEFAs, treated with 10% 3021 MRP containing mouse serum and control mouse serum, respectively, and then treated with thapsigargin. Afterward, RAW264.7 cells were co-cultured with HepG2 cells, with the corresponding controls established. The results showed that the relative protein expressions of GRP78, p-PERK, p-NF-κB and IL-1β in HepG2 cells significantly declined in 3021 MRP group compared with those in control group. However, they significantly increased in control group and 3021 MRP group after the addition of thapsigargin, without significant differences between the two groups (Fig. 5). The above findings illustrated that 3021 MRP can significantly ameliorate the abnormal activation of ERS and ROS, and inhibit the NEFA-induced inflammatory response in hepatocytes in vitro and HFD-induced inflammatory response in liver tissues of mice (Fig. 6). In addition, it was found that MRP could effectively reduce the level of inflammation and OS in mice with NAFLD, opening up a new road for research into the treatment of NAFLD in humans (Na et al., 2017). This study also had potential limitations. First, the reliability of the experimental data was lower due to a small sample size. Second, the mechanism of 3021 MRP remains to be further studied to provide a theoretical basis for further treatment of NAFLD. Last, the expressions of apoptotic factors (CHOP and cleaved-caspase-3/-8/-12) were not detected, and the effectiveness of 3021 MRP was explored only in animal models, so clinical trials should be conducted in the future. In conclusion, 3021 MRP can effectively inhibit NEFA-induced lipid accumulation and lipid toxicity in hepatocytes in vitro, restrain HFD-induced weight gain of mice, improve lipid accumulation in mouse liver tissues, and ameliorate NEFA-induced inflammatory response in hepatocytes in vitro and HFD-induced inflammatory response in mouse liver tissues through inhibiting ERS and ROS.

Supplemental Information

Supplemental Information 1 Original western blots.

Click here for additional data file.

Supplemental Information 2 Author Checklist.

Click here for additional data file.

Additional Information and Declarations

Competing Interests

Author Contributions

Animal Ethics

Data Availability

Yuanjudi Li & Zhiyun Dong are employed by Shenzhen Anxintang Biotechnology Co., Ltd. Weifang Xi is employed by Xinchen Biotechnology (Guandong) Company Limited.

Qi Xie performed the experiments, prepared figures and/or tables, authored or reviewed drafts of the article, and approved the final draft.

Shuqing Gao performed the experiments, authored or reviewed drafts of the article, and approved the final draft.

Yuanjudi Li performed the experiments, prepared figures and/or tables, and approved the final draft.

Weifang Xi analyzed the data, prepared figures and/or tables, and approved the final draft.

Zhiyun Dong analyzed the data, prepared figures and/or tables, and approved the final draft.

Zengning Li conceived and designed the experiments, authored or reviewed drafts of the article, and approved the final draft.

Min Lei conceived and designed the experiments, authored or reviewed drafts of the article, and approved the final draft.

The following information was supplied relating to ethical approvals (i.e., approving body and any reference numbers):

The animal use protocol for this study has been reviewed and approved by the Experimental Animal Ethics Committee of the Fourth Hospital of Hebei Medical University. (No. IACU-4th Hos Hebmu-2020017).

The following information was supplied regarding data availability:

The original data for the figures are available in the Supplemental File.

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
