# Peer review of "Effects of 3021 meal replacement powder protect NAFLD via suppressing the ERS, oxidative stress and inflammatory responses"

_PeerJ, doi:10.7717/peerj.16154_

## Round 0.1 · original submission · Major Revisions

A detailed point to point response letter (editor + reviewers), manuscript with tracked font are necessary for further process when they submit their revised version of paper. Please upload the response letter as a PDF file in the Supplementary materials in the online submission system. Please give detailed response. Please write down what changes have been made in the response letter, rather than reading the manuscript to find what has been revised. And please note that any uncompleted or improper corrections by the authors during this revision may lead to rejection.

The language needs revision by a fluent English speaker. Please provide a certificate of language editing service and a manuscript with tracked editing records as a Supplemental File

Reviewer 1 ·

Basic reporting

basic aim and hypothesis missing. Context not clear

Experimental design

not well stated

Validity of the findings

original data looks good .

Additional comments

In this manuscript, authors have explored the effects of 3021 Meal Replacement Powder (MRP) in obese mice. The manuscript has been written ambiguously and lack clarity and therefore is very difficult to understand hypothesis/ aims/ objectives of the study. This manuscript needs some serious improvement in terms of writing and presenting the data to be published in a journal. If done so, the manuscript has potential to educate the biologist about the advantages of 3021 MRP on liver health. I will request authors to take the criticism in good faith and try to improve the manuscript.
Comments for authors:

1. First of all, whatever information about 3021 MRP was provided in the introduction has no references. So those claims could not be verified. Please mention the references / scientific published papers which suggest that MRP regulate glucolipid metabolism and other properties ( line no 66-74 in the manuscript). lot of mistake in the manuscript like check line number 108. it states 20g of protein was loaded for western blot.
2. Abstract- there is only one line in the background which should actually be written as aim or objective. There is no actual background of the study in the background section. Please rewrite in clear words. Basically, each section of manuscript need serious makeover and needs to be rewritten.
3. The legends in the figures fail to address or explain the figures. Particularly, figure 2 is so ambiguous that I was not able to understand if the western blots should have 6 lanes or 8 lanes
4. Fig3 A suggest body weight change but I don’t see body weight parameter (gram/Kilogram/pound) etc. plotted. What is it? Please indicate what experiment/parameter has been used to show body weight changes.
5. Fig5 fails to suggest any thing.

Reviewer 2 ·

Basic reporting

1, In western blot experiments, the authors quantified the protein expression levels after compensation for beta-actin levels. In figures 2 and 4, the author should change the statistical chart to a scatter plot chart to understand the results intuitively. And moreover the total and Phosphorylated proteins for western blot should be presented together but not separated presentation. And the authors should show all blot data used in the figures as supplementary materials.
2, The numbers for each experiments must be added in the part of “results”, as state n = ? .
3, More associated references about the ERS must be added and the associated descriptions about the relationship between 3021 and ERS should be added in the part of “discussion”.
4, There are some of grammar mistakes, please find a fluent English speaker to polish this article.
5, Redraw the schematic diagram as shown in figure 6.

Experimental design

Mouse medicated serum should be added into the RAW264.7 cells as vitro experiments, mouse macropahge cell line, to identify the mechanisms found in vivo experiments.

Validity of the findings

1. exact numbers for P value shoud be added in the figure legends, and the numbers of of repeated experiments in vitro experiments should also be added in the part of "figure legends".
2. the original figures for western blot should be present as Supplementary figures.

---

## Round 0.2 · Minor Revisions

Dear Authors: I am sorry that it took so long to complete the review of your manuscript.

At this stage, Reviewer 1 asks for several items that still need to be fixed. Please make all of the revisions to ensure that your manuscript will become publishable. Thank you!

Reviewer 1 ·

Basic reporting

Authors have included the suggested changes.

Experimental design

good

Validity of the findings

valid

Additional comments

Comments for authors:
Thank you for incorporating the changes that were suggested. i will also request the authors to take my comments in positive spirit.
1.Still i find that the legend of figures do not provide sufficient information. for example in fig2. i have no idea the protein expression have been quantified from primary hepatocytes or cell culture and which cells were used . additionally, what does medivated mean and why Thapsigargin was used? you have to remember your paper will be read by people of your field and people outside your field. so paper should be self-explanatory and we should not leave even minor details.
2. In discussion, when you write about findings and conclusions about your experiments, you should write the corresponding figure number you are writing about.
Overall, the data is good and experiments are also well designed but the only problem is paper writing. it is not clearly written. if that is fixed it should be good to go

---

## Round 0.3 · Minor Revisions

Thank you for revising your manuscript. I am happy to let you know that your manuscript is almost suitable for publication in PeerJ.

During the revision, a couple of typographical errors have been introduced. Please have a fluent English speaker or professional editor proofread the text before the article goes to production.

**Language Note:** The Academic Editor has identified that the English language must be improved. PeerJ can provide language editing services - please contact us at copyediting@peerj.com for pricing (be sure to provide your manuscript number and title). Alternatively, you should make your own arrangements to improve the language quality and provide details in your response letter. – PeerJ Staff

---

## Round 0.4 · accepted · Accept

Dear Authors:

Thank you for polishing the manuscript text. Your work is now ready for publication.